# Alternating-Current Microgrid Testbed Built with Low-Cost Modular Hardware

**DOI:** 10.3390/s23063235

**Published:** 2023-03-18

**Authors:** Mark A. Haidekker, Maohua Liu, WenZhan Song

**Affiliations:** Driftmier Engineering Center, College of Engineering, University of Georgia, 597 D.W. Brooks Drive, Athens, GA 30602, USA

**Keywords:** microgrids, DC–AC inverter, power distribution, power conversion, real-time control, renewable energy sources

## Abstract

With the growing popularity of microgrids for alternative energy management, there is demand for tools that allow us to study the effect of microgrids in distributed power systems. Popular methods involve software simulation and prototype validation with physical hardware. Simulations often do not capture the complex interactions, and combinations of software simulations with hardware testbeds promise to give a more accurate picture. These testbeds, however, usually aim at the validation of hardware for industrial-scale use, which makes them expensive and not readily accessible. To fill the gap between full-scale hardware and software simulation, we propose a modular lab-scale grid model at a 1:100 power scale over residential single-phase networks with 12 V AC and 60 Hz grid voltage. We present different modules—power sources, inverters, demanders, grid monitors, and grid-to-grid bridges—that can be assembled into distributed grids of almost arbitrary complexity. The model voltage poses no electrical hazards, and microgrids can readily be assembled with an open power line model. Unlike a prior DC-based grid testbed, the proposed AC model allows us to examine additional aspects, such as frequency, phase, active and apparent power, and reactive loads. Grid metrics, including the discretely sampled voltage and current waveforms, can be collected and sent to higher-tier grid management systems. We integrated the modules with Beagle Bone micro-PCs, which in turn connect any such microgrid with an emulation platform built on CORE (Common Open Research Emulator) and the Gridlab-D power simulator, thereby allowing hybrid software/hardware simulations. Our grid modules were shown to fully operate in this environment. Through the CORE system, multitiered control and even remote grid management is possible. However, we also found that the AC waveform poses design challenges that require us to balance accurate emulation (most notably with respect to harmonic distortion) with per-module costs.

## 1. Introduction

With growing support of renewable energy sources, microgrid installations become more and more widespread. One assumption is that microgrids can take some load off the power distribution system and thereby reduce power costs and environmental impact [1]. However, the increasing number of local microgrids with the ability to feed power into the main grid can cause challenges, such as grid management and optimization, especially in light of a continuously changing topology, as microgrids connect and disconnect based on their power production [2]. Even more consequential, the main grid infrastructure must have the capability to bridge periods of low local energy production due to the intermittent and unreliable nature of microgrid energy production. Without this additional baseload capability, there exists an increased risk of cascading blackouts that can cover wide areas [3]. With these developments, the management of complex, interconnected grid sections, which are often referred to as “smart grids”, becomes an urgent research area [4,5].

A common practice to test smart grid concepts is by means of simulations [5,6,7]. However, smart grids consist of multiple actors and physical phenomena that are often difficult to capture in one single simulation framework. Therefore, researchers increasingly couple distinctly different simulators to form more encompassing “co-simulations” [8,9,10,11]. Yet, some real-world applications require prototype implementations on actual testbeds with physical hardware to validate simulations and test concepts before deployment in the power systems industry and for public use. Such testbeds have been developed previously (see, for example, [12,13,14,15,16]), but their focus is grid-scale analysis of their performance, which makes them expensive and not readily accessible. Recently, lab-scale testbeds were introduced [17,18]. Our group also presented a lab-scale hardware testbed, referred to as SmartGridLab [19], which provides a low-cost means to design heterogeneous grid infrastructure with solar panels, wind turbines, smart appliances, energy storage, and facilities to route energy to different grid sections [20,21].

The main contribution of this paper is to present the first prototype of a low-voltage and table-top AC SmartGridLab testbed, in which four distinct modules (power sources, power demanders, power meters, and grid-to-grid switches) can be combined into microgrid systems of almost arbitrary complexity. The goals of the present AC microgrid are (1) to advance our previous DC SmartGridLab [19] by enabling alternating current grids, which imposes additional aspects of frequency and phase and thus allows the study of reactive power, reactive loads, and frequency disturbance; (2) to implement a three-tiered hierarchical control that ranges from control on the module level with the capability to combine and control multiple modules on the subgrid level, and reaches the integration of subgrids with a simulation and management system that was built on CORE (Common Open Research Emulator) and the Gridlab-D power simulator; and (3) to keep the module costs as low as possible to allow upscaling of the grid models within typical classroom budgets.

## 2. Materials and Methods

We decided to design a single-phase grid at a 1:100 scale from the consumer-level 120 V utility, i.e., a grid with 12 *V*${}_{rms}$ and 60 Hz with the current scaled down proportionately. A 1 kW consumer, such as a resistive heater, would be represented by a 10 W model consumer, and a 10 W model solar cell would represent a 1 kW solar installation. The voltage of 12 V AC does not pose any electrical hazards and is therefore safe to use in a lab setting. The model for the main utility grid was implemented with a standard 12 V, 5 A transformer. The four elements of the DC SmartGridLab (power source, power demander, power meter, and grid-to-grid switch) find their representation in the inverter, demander, monitor, and bridge, each of which are described in detail below. Each of these modules has in common an internal 60 Hz oscillator, which is synchronized with the AC grid by means of a phase-locked loop (PLL), and an I2C communications interface, which allows each module to be monitored and controlled from a higher-level computer, which would typically be represented by a mini-PC such as the Raspberry Pi or BeagleBone Black. An overview of the relevant components in a possible grid/subgrid configuration is shown in Figure 1.

### 2.1. General Design Considerations

Each module requires its own dedicated microcontroller. We decided to use 8-bit RISC microcontrollers of the Microchip PIC18F series. The use of such a fairly low-end microcontroller reduces the per-unit costs. At a sampling rate of 3.84 kHz, and with the use of integer arithmetic, the microcontroller’s performance is sufficient to complete all required operations in real time. We also made use of the microcontroller’s built-in 10-bit analog-to-digital converters (ADCs) and provided analog circuitry (amplification and level conversion) as needed for each module. The microcontroller and its analog amplifiers are strictly isolated from the AC grid by means of optoisolators to avoid any ground currents through the high-level control or monitoring components. To create compact and robust units, we developed circuit boards for each of the modules. Each module has screw terminals for connecting it to the grid. In addition, terminals or connectors exist for the 5 V lab power supply and for the I2C bus. Although small-sized microgrids can be assembled from the PCBs, we also developed 3D-printable cabinets. The cabinets feature matching connectors for lab power, I2C, and grid connectivity.

We decided to use the I2C bus, because it allows addressable devices and because I2C functionality is available on micro-PCs, such as the BeagleBone Black and the Raspberry Pi, even though the bus connection runs across several boards. The I2C specifications limit the bus length to 10 m, and a typical subgrid I2C cable in our setup is no longer than 50 cm of shielded cable. In addition, we used I2C level converters (PCA9306, Texas Instruments) to adapt the BeagleBone’s 3.3 V level to the control board’s 5 V level, and bus traces between connector and the PCA9306 IC were kept very short. We used a standard-mode speed of 100 kHz with pull-up resistors of 2.2 kΩ. Data were transferred as 16-bit integer values, and scaling occurred on the micro-PC.

### 2.2. Grid Sensing and Phase-Locked Loop

Each module features an identical circuit to measure the grid voltage, grid current (i.e., current delivery by an inverter, current draw of a demander, or current between two subgrids), and phase. The grid voltage is scaled down by a factor of 120 with a resistive voltage divider and fed into a Si8920B isolation amplifier. The current is measured across a 0.1 Ω shunt resistor and fed directly into a second Si8920B isolation amplifier. Since the Si8920B allows ±200 mV at its input, the line voltage can reach up to a ±24 V peak, and the current is allowed to reach a ±2 A peak. On the isolated side, the differential outputs of the Si8920B are offset-adjusted and amplified by a factor of 1.5. Together with the 8.1× amplification of the Si8920B, the grid voltage is scaled by a factor of 0.1 (i.e., ±24 V on the grid side translates into 2.5 V ± 2.4 V on the microcontroller side). The current is scaled by a factor of 1.2 V/A (i.e., ±2 A to or from the grid side translate into 2.5 V ± 2.4 V on the microcontroller side). With the internal 10-bit ADC of the microcontroller, the resolution limit is 50 mV/LSB for the grid voltage and 4 mA/LSB for the current. The worst-case power deviation is therefore 20 μW for a single measurement or 5.2 nJ for a measurement at the 3.84 kHz sampling rate.

Each module also features a phase-locked loop (PLL), which is driven by the discrete zero-crossing event of the AC line. The zero-crossing event is derived from the output of the Si8920B voltage isolator in a comparator with a 1.6 mV hysteresis. The resulting phase lag is 0.001 · π. At each zero crossing, an interrupt signal activates the PLL control software, which takes the form of a time-discrete control system that synchronizes the internal oscillator of the microcontoller with the AC grid. The internal oscillator is also responsible for producing the 3.84 kHz sampling frequency, which is used (a) to generate a sampled sine signal with a software-controllable phase shift, (b) to control the analog acquisition of the momentary AC grid voltage and current, and (c) to perform task scheduling. A schematic overview of the PLL is shown in Figure 2.

The time between zero-crossing events is measured with one of the microcontoller’s internal timers, and the phase difference Δφ emerges as the difference between the internal oscillator timer and the zero-crossing timer. The control action is computed after every zero crossing, i.e., with a sampling rate of T=8.33 ms. The controller is a time-discrete PI controller with the *z*-domain transfer function
(1)$$H\left(z\right)=\frac{\left(k_{p}+k_{I}T\right)z-k_{p}}{z-1},$$
where *T* is the sampling rate of 8.33 ms and $k_{p}$ and $k_{I}$ are the controller gain coefficients. The control target is the internal oscillator’s phase $\varphi_{OSC}\left(z\right)$, which follows the external grid phase $\varphi_{AC}\left(z\right)$ and the reference frequency $f_{0}=60$ Hz through
(2)$$\varphi_{OSC}\left(z\right)=\frac{2\pi g_{0}TH\left(z\right)}{\left(z-1\right)+2\pi g_{0}TH\left(z\right)}\varphi_{AC}\left(z\right)+\frac{2\pi Tg_{0}}{\left(z-1\right)+2\pi g_{0}TH\left(z\right)}\cdot \frac{z}{z-1}f_{0}$$
where $g_{0}$ is a unitless coefficient that combines the PLL timer resolution and the linearized gain of the reciprocal control action, i.e., the corrective action that is a change of the oscillator’s period register. If the line voltage is absent, the internal oscillator runs with the frequency $f_{0}$ of 60.000 Hz. A higher-level control device (host PC) can read the oscillator’s period register and obtain the PLL frequency with a precision of 0.01%.

### 2.3. Inverter

The core of the inverter is a pulse-width modulated (PWM) H-bridge with an LCL output filter. The basic schematic is shown in Figure 3. The component groups of D2, S2, L2 and D1, S4, L1 form switch-mode buck converters, whereby the switches S2 and S4 are MOSFETs. The corresponding half-wave switches S1 and S3 (also MOSFETs) are turned on during the positive and negative half-wave, respectively, and the associated buck switch receives the pulse-width modulation (PWM) signal. The inverter’s control software establishes the AC grid’s phase and amplitude and controls the H-bridge to feed a sinusoidal current into the grid that is proportional to the AC voltage.

Inverters usually employ a feedback scheme for controlling the grid current [22], but when several assumptions are made, the pulse width can be computed beforehand. This principle is sometimes referred to as “predictive control” [23,24]. The grid voltage $V_{g}\left(t\right)$ is sinusoidal (Equation (Equation 3)). To generate active power, the current $I_{g}\left(t\right)$ from the inverter into the grid has the same phase and frequency,
(3)$$\begin{array}{c} V_{g}\left(t\right)={\hat{V}}_{g}\cdot \sin\left(\omega t\right)\\ I_{g}\left(t\right)={\hat{I}}_{g}\cdot \sin\left(\omega t\right) \end{array}$$
where ${\hat{V}}_{g}$ is the peak voltage of $12\cdot \sqrt{2}$ V ≈17 V and ω=2π·60 Hz = 377 s^−1^. The peak grid current, ${\hat{I}}_{g}$, is controlled by the inverter and depends on the DC-link voltage. The PWM switching frequency $f_{PW}$ is significantly higher than the grid frequency. Hence, the input power follows the output power instantaneously (conservation of energy). Therefore, the input current, averaged with respect to the PWM frequency, but momentary with respect to the AC grid frequency, becomes
(4)$$I_{in}\left(t\right)=\frac{I_{g}\left(t\right)\cdot V_{g}\left(t\right)}{V_{DC}}$$

If the filter capacitor C1 is large enough, we can assume that the voltage across C1 is identical to the grid voltage $V_{g}\left(t\right)$. Since $I_{in}$ is the average of the inductor current during the PWM on-time and $V_{e}=V_{DC}-\left|V_{g}\right|$ is the voltage across the inverter-side inductor (L1 or L2), we can use the inductor equation to eliminate $I_{in}\left(t\right)$ and solve for the pulse width $p=\tau \cdot f_{PW}$:(5)$$p\left(t\right)=\left.\sqrt{Lf_{PW}\frac{P_{out}}{V_{DC}}}\;\cdot \;\frac{\left|\sin(\omega t)\right|}{\sqrt{V_{DC}-{\hat{V}}_{g}\left|\sin\left(\omega t\right)\right|}}\right.=\alpha \;\frac{\left|\sin(\omega t)\right|}{\sqrt{1-\beta |\sin(\omega t)|}}$$
where $P_{out}$ is the effective output power, i.e., $P_{out}=V_{g,\;eff}\cdot I_{g,\;eff}={\hat{V}}_{g}{\hat{I}}_{g}/2$. The equation can be simplified by combining several constants into two coefficients, α, which controls the amount of power fed into the grid, and β, which depends on the DC-link voltage. In this form, discrete values for *p* can be computed beforehand and stored as a table, which significantly reduces the computational effort. In practice, the coefficient α serves two additional functions. First, the coefficient may be attenuated from 0 to 100% of its theoretical value through a higher-level control process. Second, adjusting α can be used to adjust the output power component to the DC-link voltage excess $V_{e}$.

The bridge MOSFETs (S1 through S4 in Figure 3) are driven by the microcontroller through digital optoisolators. A mechanical relay ensures that the inverter is connected to the grid only under nominal operating conditions. A shunt resistor (0.1 Ω) in the output path allows to measure the current $I_{g}\left(t\right)$. Both current and voltage signals are sent to the microcontroller through isolation amplifiers. The different interacting sections of the inverter can be seen in Figure 4.

The inverter module allows two distinct modes of operation. First, it can be used in an isolated mode where the internal oscillator runs unsynchronized at 60 Hz. In isolated mode, the inverter simply converts power from the DC-link to its AC output, whereby the maximum power can be controlled either from the DC-link or by software. If the inverter is connected to any grid that carries 60 Hz line voltage, the inverter senses frequency and phase and synchronizes to the line voltage before connecting. The assumption in this case is that the grid acts similar to a stiff bus bar and is not significantly influenced by the inverter’s power contribution.

In the second mode, multiple inverter modules in a subgrid can be controlled from a grid-to-grid bridge. This mode is activated when a bridge module is connected to the inverter (connector labeled “subgrid control” in Figure 4), in which case the bridge module provides the reference phase and controls whether or not the inverter is allowed to connect to the subgrid, while the inverter can signal phase lock to the bridge.

### 2.4. DC–DC Converter and Power Tracking

The inverter requires a stable DC-link voltage $V_{DC}$, which must be larger than the grid peak voltage. Renewable power sources (e.g., solar cells, wind turbines) generally do not meet this condition, and we provided a DC–DC boost converter to generate the DC-link voltage. The topology is indicated in Figure 1. The DC–DC boost converter is built around a MC34063 switch-mode controller in boost configuration. The converter is powerful enough to boost the power from the 10 W (nominal) solar cell that was used in the DC SmartGridLab. In addition, the DC–DC converter was used to realize a form of maximum power tracking. The DC-link voltage is made variable within limits: the allowed range is 23 V to 30 V. A feedforward scheme adjusts the boost converter output voltage depending on the input voltage, and the DC-link voltage characteristic can be described as
(6)$$V_{DC}=\left\{\begin{array}{cc} 12V\; & \mathrm{for}\;V_{in}<7.5\;V\\ 1.5\cdot V_{in}\; & \mathrm{for}\;7.5\;V\leq V_{in}\leq 18.75\;V\\ 30V\; & \mathrm{for}\;V_{in}>18.75\;V \end{array}\right.$$

When the DC–DC converter receives solar power and provides it to the inverter, the inverter’s output power is adjusted based on the DC–DC converter’s output. As the inverter draws current from the DC-link, the current drawn from the solar cell increases and its output voltage decreases. This output voltage drop then signals the inverter to reduce its grid power. Combined, this scheme creates a low-gain voltage feedback, which keeps the output voltage of the solar cell close to its maximum. This operating point is similar (but not identical) to the operating point determined by maximum power point tracking (MPPT), and because our feedback scheme attempts to keep the solar cell voltage near its maximum, we termed it voltage priority power tracking (VPPT).

For laboratory demonstration purposes, we designed a variation of the DC–DC converter in which the solar cell or wind turbine provides the VPPT control signal only, and the actual power is drawn from a lab power supply. A second variation uses three lithium-ion rechargeable batteries (18,650 type) as power source. A battery charger draws power from the AC grid and charges the batteries when AC grid power is present. If the primary AC grid power fails, the DC–DC converter is enabled, and battery power is supplied to the inverter. This second variation is the equivalent of the DC SmartGridLab’s energy storage module.

The DC–DC converter is shown in Figure 4. Figure 4 also highlights how the DC–DC converter is used in conjunction with the inverter module, and the different configurations as solar/wind converter and as storage unit are shown.

### 2.5. Demander

The demander was designed around a power transistor in emitter follower configuration, in which the voltage drop across the emitter resistor (0.33 Ω) is fed back through an operational amplifier into the base. In this fashion, the collector current is approximately proportional to the voltage at the positive input of the operational amplifier divided by 0.33 Ω, and the transistor acts as a programmable current sink. The AC signal is rectified with four Schottky diodes, and the same transistor provides the programmable load for both half-waves. A microcontroller generates the sinusoidal control voltage with its PLL. The amplitude of the signal (and thus the peak load current) can be set via I2C, and the phase can be influenced as well (Figure 2). However, the load is purely active. Emulation of reactive loads is only approximate, because out-of-phase current cannot be fed back into the grid. In addition, the rectifier diodes have a small forward voltage (combined approximately 0.4 V), and no current flows near the zero crossings. A power factor of unity cannot be achieved with this circuit; however, the present demander has a low complexity and low cost.

### 2.6. Grid Monitor

The monitor, shown in Figure 5, contains the optoisolators and amplifiers to measure the AC grid voltage and the current that flows through a small shunt resistor. A monitor can be placed anywhere between two subgrids or between the transformer and a grid. It features a display, which allows instant monitoring of the grid conditions, such as a power drop from the transformer when an inverter begins operating and adds power to the grid. The monitor also allows sampling of the grid voltage and current at the 3.84 kHz sampling rate, which allows the higher-level control software to read the sampled AC wave and to compute, for example, harmonic content or harmonic distortion. One monitor is an integral part of the transformer unit.

### 2.7. Grid-to-Grid Bridge

The grid-to-grid bridge, sometimes also referred to as switch, is a bidirectional solid-state relay, which allows to connect two subgrids and therefore allows power to flow in either direction. A control circuit allows the bridge to close only when both subgrids have the same frequency and phase. The latter requirement is the most significant difference to the switch module of the DC SmartGridLab. The AC bridge, therefore, is mainly intended to control a subgrid and prepare it for energy delivery to the main grid or to its hierarchical higher-level grid. For this purpose, its PLL locks onto the main grid. The bridge module can assert control over any connected subgrid inverters and move them toward phase synchronization with the main grid, upon which the bridge connects the subgrid and main grid. Apart from the inverter control section, its circuit is very similar to that of the grid monitor. Figure 5 shows the bridge module and highlights its relationship to grid, subgrid, and inverters.

The bridge communicates with one or more attached inverters of a subgrid through four additional control signals. The request to establish a connection between the main grid and the subgrid is initiated by Tier-2 software. When this request is issued (and the main grid carries a defined 60 Hz line voltage), the bridge ensures subgrid synchronization in the following sequence:The main grid’s frequency and phase are applied to the signal PHASE.The bridge activates the signal SUBGRID_REQ which forces the inverters to disregard the AC line voltage and directs the inverter’s PLL to lock on to the bridge’s PHASE signal.Each inverter that achieved phase lock releases the open-collector line LOCKED, which becomes high once all inverters are phase-locked. At this point, the inverters are allowed to close their line relays, but keep the delivered power at zero.When all inverters are phase-locked, the bridge closes its grid-to-grid relay and signals this state to the inverters through the SUBGRID_CONN signal. In this state, the inverters begin to linearly ramp up their inverter bridges by a constant increment per AC cycle until they reach their maximum power that is dictated by the DC-link excess voltage.

The subgrid can be disconnected from the main grid, also by Tier-2 software. The bridge initiates a ramp-down of the inverter power by releasing the SUBGRID_CONN signal and opens its grid-to-grid relay after a constant wait period. Once the grid-to-grid relay is opened, the signal SUBGRID_REQ is deactivated and the inverters move into an idle state.

### 2.8. Integration of the Hardware Modules with the Emulation Platform

The modules of the AC grid testbed were designed to be integrated with a Smart Grid software emulation platform. The software platform is based on the Common Open Research Emulator (CORE, [25]) and the Gridlab-D power simulator [26]. The overall architecture of CORE consists of a graphical user interface (GUI) and a software emulator. The GUI allows remote access to the software emulator, which contains—relevant for this project—a service layer that converts GUI requests into Ethernet socket events that are forwarded to micro-PCs (BeagleBone Black). The micro-PCs then communicate with the attached grid hardware modules to collect data or to configure them. Figure 6 shows a schematic overview of the flow of information from the user front-end to the hardware modules.

Two software components of the emulation platform are InfluxDB (data collection) and Grafana (data visualization). Data, such as $V_{rms}$, frequency, power, and discrete waveform data, are queried from the micro-PCs and managed with InfluxDB. InfluxDB then acts as a data source for Grafana, whose output can be viewed at the GUI front-end.

### 2.9. Energy Sources and Measurement Conditions

We used the same model solar cell and wind turbine that we used in the DC SmartGridLab model. The solar cell was a 10 W model (STP010P, UL-Solar Inc.) with a maximum power point at 17.6 V. The model wind turbine was custom-manufactured from a 200 mm computer fan, in which the motor control was circumvented and the coils routed to the terminal wires through a bridge rectifier. Thus, the permanent magnet that rotates with the fan blades induces a weak current in the coils, which is made available at the terminal wires.

Unless otherwise specified, voltage, current, and power measurements were performed in a grid–subgrid configuration, in which the model utility power was provided by a 12 V, 2.5 A transformer (no-load voltage of 14 V) and fed into a 20 Ω resistor as the load.

## 3. Results

To test the performance of the grid testbed modules, we built two model grids. One model consisted of one transformer, two inverters with associated DC–DC converters, one demander, and a resistive load (12 V/20 W halogen lamp). The second model was similar, but also contained one bridge module. A simple visual and electrical representation of the grid itself was built from an aluminum electrical breadboard (earth) and an elevated copper wire (live). Modules could be attached with alligator clips. One such system is shown in Figure 7. In this figure, a grid–subgrid setup can be seen that follows, to some extent, the sketch in Figure 1. Two live wires are mounted atop an aluminum board (Figure 7B): one for the utility grid and one for the subgrid. Both are connected with a bridge (Figure 7C), and the utility wire is fed from the transformer box (Figure 7A). The subgrid receives power from the solar cell via DC-link converter and inverter (Figure 7E–G) and from a storage unit (Figure 7H,I). A gray ribbon cable is visible that allows control of the inverters from the bridge. Additional wiring includes the 5 V lab power for the microcontrollers and the I2C bus. One demander is visible, and a resistive load is placed on the aluminum board. The modular design allows to set up a variety of different subgrid topologies.

### 3.1. PLL Performance

Since the PLL is shared by all modules, we tested the PLL separately. We found experimentally that the capture range is in excess of 60 ± 20 Hz, which allows the PLL to lock even under extreme mismatch conditions. If we defined the tolerance band for the PLL phase as 0.005 π, which corresponds to a deviation of the zero-crossing events between external AC signal and the internal oscillator of approximately 42 μs, we found the PLL to settle inside the tolerance band within less than 120 ms. One example measurement where the PLL was forced to synchronize to the 60 Hz AC frequency from an initial free-run frequency of 40 Hz is shown in Figure 8.

### 3.2. Inverter Performance

The inverter, as presently configured, can deliver up to 15 W into a stiff bus bar, i.e., an AC grid whose voltage is not significantly influenced by the inverter’s current. In one test configuration, the grid was loaded with 10 W (resistor of 15 Ω, visible on the board B in Figure 7), and the inverter provided 5 W. The output waveforms are shown in Figure 9. A limited amount of harmonics is introduced, in part due to integer rounding effects, and in part due to nonideal properties of the LCL filter, such as copper resistance and capacitor ESR. Harmonic distortion increases as the inverter’s output power decreases. The harmonics are a trade-off of the feedforward (“predictive”) control scheme, but the advantage is its reduced cost due to the absence of the feedback control system and increased robustness, since potential instabilities and LCL resonance effects are avoided.

The selected microcontrollers were adequate for the computational needs. In the two-tiered interrupt system, the oscillator interrupts and zero-crossing interrupts were assigned to the high-priority level. The oscillator interrupt initiates data acquisition, although the readout of the ADC is serviced by a low-priority interrupt. Both high-priority interrupt service routines (ISR) and the ADC readout ISR are executed within 3–4 μs each. Analog data acquisition is sequential (grid voltage, inverter-to-grid current, DC-link voltage, reference voltage), and the conversion (12 μs duration) takes place in the background. Total analog conversion time including basic data processing is, therefore, 55 μs out of 260 μs sampling rate. Each conversion triggers a non-interrupt processing function for sample summation and peak detection (approximately 20 μs duration). The PLL computations take place after each zero crossing (outside of the ISR) and take 52 μs, most of which is executed in parallel to the ADC for the first sample. The final set of computations is scheduled after the last ADC in each full wave (duration of 150 μs, which includes three calls of an integer square root function) in which the active power *P* and the RMS voltage and current are computed according to the well-known formulas
(7)$$V_{rms}=\sqrt{\frac{1}{N}\sum_{k=0}^{N-1}V_{k}^{2}}\;\;;\;I_{rms}=\sqrt{\frac{1}{N}\sum_{k=0}^{N-1}I_{k}^{2}}\;\;;\;P=\frac{1}{N}\sum_{k=0}^{N-1}V_{k}\cdot I_{k};$$
where *N* is the number of samples per full wave (N=64 in our case), and $V_{k}$ and $I_{k}$ are the discrete samples taken at the sampling frequency of 3.84 kHz. Under the definition of Equation (Equation 7), the active power of the inverter is negative to indicate power fed into the grid.

The power tracking behavior of the inverter supplied by a solar cell and a DC–DC converter is shown in Figure 10. The output voltage of the DC–DC converter rises with increasing solar cell output voltage (Figure 10A), and the inverter begins feeding current into the grid when the DC-link voltage $V_{DC}$ exceeds 22 V. The peak current is proportional to the voltage excess $V_{DC}\text{—}22$ V. A voltage hysteresis function prevents the inverter from rapidly turning its bridge on and off; the action of the hysteresis can be seen in Figure 10B, in which the current drawn from the DC-link drops close to zero before the bridge is turned off. Correspondingly, the inverter first allows the solar cell voltage to rise to a level close to its peak power point before increasing the power output. At this point, the inverter increases its output power to a point that the cell voltage remains close to the peak power voltage (Figure 10C). The data points were taken from a time series with randomly fluctuating solar irradiation.

The overall efficiency of the inverter combination is approximately 55%, but the efficiency can be as low as 40% for solar power at the lower limit of the DC-link voltage. We were able to show, however, that a 12 V AC/6.5 W LED replacement lamp for low-voltage halogen spotlight applications was brightly lit while it was powered exclusively and autonomously from a solar cell and the inverter combination.

A rough estimate of the solar cell’s efficiency was made under unoccluded sunlight with 77,000 lx. Based on the geographic location, we assumed the sunlight’s efficiency to be 80 lm/W, and the solar cell can be expected to receive 960 W/m${}^{2}$. This is close to the test conditions of 1 kW/m${}^{2}$. With an area of 32 × 25 cm, it receives approximately 77 W. Under this illumination, the cell delivers 250 mA with a voltage of 16 V (4 W), and its degree of efficiency can be estimated to be near 5%. Under lab fluorescent light of 7000 lx (luminous efficiency approximated with 60 lm/W), the efficiency of the solar cell drops to 0.45%, and the cell provides a power of only 40 mW. The model wind turbine produces a time-varying voltage that looks similar to a rectified AC voltage with an open-circuit peak of 5 V. Driven at 600 rpm, it can provide 14 mA into a 220 Ω resistor, but its peak voltage drops to 3 V. Therefore, its output power is similar to that of the solar cell under indoors illumination.

Both the DC–DC converter and the inverter require power to operate the internal control circuitry. At nominal solar cell conditions of 16 V cell voltage and 28 V DC-link voltage, the power requirement is 0.8 W, which has to be supplied by the power source. Neither the model wind turbine nor the solar cell under indoors lighting conditions is capable of satisfying the internal power requirement. For this reason, we designed a variation of the DC–DC converter with a weak load (resistor of 470 Ω) for the solar/wind source, whose voltage is used only to control the DC-link voltage. The supply for the boost converter is provided by a lab power supply (Figure 7). In a lab setting, the inverter behavior with this form of amplified DC–DC converter is intuitive in that varying light levels and varying wind turbine speeds cause the inverter to accordingly adapt its power generation. A simulation of, for example, the sudden addition of a power source to the grid or a partial blackout is therefore possible.

### 3.3. Demander and Monitor

The demander module can be controlled by the higher-level software, and we found a linear relationship between the control data word and the current drawn (6 mA/lsb, $R^{2}>0.999$) and the power demand (71 mW/lsb, $R^{2}>0.999$). However, the grid-side control circuitry draws 120 mA even when the control word is zero. Moreover, the low-cost principle of using the rectified AC voltage and a single-transistor current source with a controlling operational amplifier causes noticeable current spikes near the zero-crossing points at φ=0° and φ=180°. These spikes influence the grid voltage (Figure 11). For small phase shifts (45° shown in Figure 11B), the distortion does not significantly increase, but larger phase shifts (90° shown in Figure 11C) cause major harmonic distortion, mainly due to the absence of a true reactive component (i.e., where an inductor or capacitor would return stored energy to the grid). In its present form, the demander is therefore limited to small phase shifts between grid voltage and demander current.

The monitor displays real-time data without the need for additional hardware or software (LCD visible in Figure 7). It displays grid voltage (peak-to-peak and rms, alternating), current through the monitor’s shunt resistor, active power *P* and apparent power $S=V_{rms}\cdot I_{rms}$, and the power factor, computed as |P|/S (see Equation (Equation 7)). These values can be queried by higher-level control software, but the higher-level control systems can also recall the sampled grid data $V_{k}$ and $I_{k}$ over one full period. The sampled data are continuously updated, but update is halted during data transfer and only data from the last complete full wave are transmitted. These data can be used to compute the harmonic distortion or the phase angle imposed by the load. Examples of such waveforms are shown in Figure 12 with (A) a load resistor, (B) an inductor of 31 mH and 1.6 Ω wire resistance, (C) an electrolytic capacitor of 120 μF, and (D) a nonlinear load, i.e., a 12 V AC, 6.5 W LED lamp with built-in rectifier and switch-mode current control. The power factors reported by the monitor were 0.98, 0.18, 0.04, and 0.92, respectively. The inductor would have a phase angle of 82° and therefore a theoretical power factor of 0.13. The reported value is close, and additional resistances (shunt, contacts) may account for some of the difference. The irregular waveform in Figure 12C may be attributed to LC resonance effects between the transformer secondary coil and the capacitor.

### 3.4. Grid-to-Grid Bridge

Unlike in a DC grid model, where power routing can be accomplished with a simple relay or solid state switch, the AC grid bridge must ensure matching frequency and phase between subgrids. Therefore, the bridge module needs to assume control over the subgrid inverters. We provided an alternative PLL input for the inverters through which the bridge module can provide the frequency reference and through which the inverters can signal when they are phase-locked. In its practical realization, the following sequence of communication signals was implemented: After the higher-level control software signals the bridge to close, the bridge begins sending the frequency reference to the subgrid inverters. Each inverter signals readiness when it has locked onto the frequency reference. The bridge then closes the electrical link between the two subgrids and sends a corresponding signal to the inverters. Closing the bridge’s solid-state relay takes place at a zero crossing. The inverters then enable their power stages and begin feeding power into the subgrid and, through the bridge, into the grid. Conversely, when the higher-level software instructs the bridge to open, the inverters receive a signal to cease feeding power into the grid, after which the bridge opens the connection. Closing the bridge and waiting for inverters to achieve phase lock can take several seconds, and the higher-level software must query the bridge status.

In practical operation, we found that transients can occur when the bridge establishes a grid-to-grid connection, especially when reactive loads are present. These transients occasionally cause the inverters to drop out of phase lock, which is an error condition that is signaled back to the bridge. To alleviate this problem, we allow the switch to ignore the inverter’s status for several 60 Hz cycles after it closes. However, the zero-crossing PLL has marginal noise immunity for more complex grid structures.

### 3.5. Communication with Micro-PC

The individual module’s communication with the BeagleBone micro-PC that is responsible for second-tier control warrants separate examination. For the standard-mode clock frequency of 100 kHz, a rise time of no more than 1 μs is required. Each PCA9306 has approximately 11 pF input capacitance, to which is added the cable capacitance of 30 pF for a signal line to ground. In a typical subgrid with five bus devices, the total capacitance for a signal line was 85 pF, which gives a theoretical rise time of 0.2 μs (see, e.g., Texas Instruments Application Report SLVA689). We verified with the oscilloscope that the rise time was clearly below 0.3 μs. The short rise time would even allow 400 kHz fast mode communication, but we kept the I2C bus on standard-mode speed to maintain a safety margin. Communication failures, such as I2C bus lockups, were not observed.

With standard-mode speed, a 16-bit register read takes approximately 0.5 ms. Since data are transferred as 16-bit integers, the read-out of grid voltage, current, and power (Equation (Equation 7)) takes about 1.5 ms, which is far less than the AC half-period of 8.33 ms. When a monitor’s sampled AC voltage and current are collected, a transfer of 256 bytes is necessary (64 samples per cycle, 16 bits per sample, each for voltage and current). The transfer takes about 14 ms each for voltage and current. Only one such transfer can be performed for a full AC cycle, and buffering ensures that the data are collected during the same cycle. However, analysis of the waveform is optional and infrequent within the context of monitoring power distribution.

### 3.6. Case Study: Management of Intermittent Renewable Energy

We assembled a grid model with two inverters, a 20 W resistive load, and a transformer. The inverters were connected to a solar cell and a model wind turbine, respectively. The grid modules were connected to a BeagleBone micro-PC and, in turn, to the CORE software system. The purpose of this case study was to show how the power grid provides stable output despite the intermittent nature of the renewable energy sources.

The three power sources (solar panel, wind turbine, and transformer as model for the wider grid infrastructure) were connected to one grid and fed into the 20 W load. The power output of each power source was plotted by Grafana and is shown in Figure 13. Initially, both renewable sources were active, and the transformer (i.e., the larger utility grid) supplied only the power difference. The initial power output of the solar panel was about 3.7 W, the power output of the wind turbine was about 2.6 W, and the transformer supplied the difference of 13.7 W. In the course of the experiment, one or both of the renewable sources underwent a simulated failure, but the output remained constant at 20 W, because the shortfall was supplied by the transformer. These events can readily be seen in Figure 13.

## 4. Discussion

With this project, we demonstrated the feasibility of a model grid for AC power. The modules that can be used for different grid configurations are equivalent to those of the DC SmartGridLab [19] that served as our reference: power source (i.e., inverter with DC–DC converter), demander, monitor, grid-to-grid bridge, and storage. We consider the testbed hardware low-cost, because the component costs for one subgrid set (in this example, two inverters with associated DC–DC converters, one demander, one monitor with transformer, and one bridge) add up to less than USD 150, which includes printed-circuit boards, but not the 3D-printed cabinets. The controlling BeagleBone Black micro-PC adds another USD 55. Careful sourcing (e.g., surplus components) can further reduce the costs. Models for renewable energy sources are counted separately, but under consideration of (a) the nature of the wind turbine, which is a converted PC case fan, and (b) the amplified DC–DC converter that provides sufficient “solar” power even under poor lighting conditions, it can be assumed that low-cost solutions can be found without difficulty.

Compared with the DC SmartGrid Lab testbed, we encountered several challenges that were imposed by the AC nature of the model grid. The DC SmartGridLab uses, by default, the built-in 5 V lab power source, and energy inputs are merely control inputs, in analogy with the amplified DC–DC converter in the present project. Since the model grid carries 5 V DC, the dynamic response of grid modules is not a critical consideration. For example, the DC SmartGridLab’s energy source has a built-in time constant of 10 kΩ · 1 μF = 10 ms, which represents more than 180° in a 60 Hz AC system. The time constant is caused by a typical frequency compensation RC network used in high-gain operational amplifier circuits. The AC demander uses a similar circuit with a time constant of only 1 μs, yet the feedback phase lag causes significant spikes in the demander waveform (Figure 11A). For the AC grid design, the dynamic response is, therefore, a critical consideration. For the same reasons, the accuracy of the internal oscillator of the modules is equally critical. Near the zero crossing, the 12 AC voltage rises by approximately 6.4 V/ms, and even microsecond-duration timing errors cause significant errors in the measured grid metrics. Our PLL allows a maximum phase error of 0.9° (80 μs between AC line and PLL oscillator zero crossing), which we considered to be an acceptable compromise between PLL accuracy and sensitivity.

In fact, the requirement of a PLL is another fundamental difference between the DC SmartGridLab and our present AC grid testbed. The PLL in its present form is based on the discrete zero-crossing events of the AC line, which are straightforward to detect with an analog comparator, and computationally inexpensive, because the PLL algorithm needs to run only every 8.3 ms. However, the zero-crossing detector is relatively noise-sensitive. This sensitivity is known, and remedies have been proposed [27]. Specifically for our modules, transients near 0° and 180° can cause significant phase errors. This was observed especially with the bridge module. Out of several possible solutions [27], we see a PLL formulated as a 60 Hz bandpass filter [28] as an attractive alternative. The filter produces a quadrature pair of signals of amplitude $\hat{W}$. An error signal ϵ(t) can be defined between the in-phase component of the quadrature oscillator and the grid voltage:(8)$$\epsilon \left(t\right)={\hat{V}}_{g}\sin\left(\omega t\right)-\hat{W}\sin\left(\omega_{0}t+\varphi \right)$$

When the grid voltage serves as the input of the bandpass filter, its output follows the input both in phase and amplitude, thereby minimizing the time-integrated error ϵ(t) [28]. Due to the bandpass nature, however, transients are effectively suppressed. Computational effort for such a filter is significantly higher than the present solution, but the advantage is that no additional hardware effort is needed. Moreover, this PLL algorithm can be readily implemented with the discrete grid samples of our system (Equation (Equation 7)).

Another critical difference to DC grid models is the need for full isolation between the AC grid-side components and the control components. This isolation is needed, because most PC components are grounded, and this ground connection propagates through communication cables. While it would be possible to define the AC neutral as ground, it would severely limit the flexibility of component placement (e.g., shunt resistors). Moreover, the renewable energy sources *must* be isolated due to the inverter H-bridge. Apart from the costs, the optoisolators require power, typically tens of mA, which needs to be provided either by the grid or the power sources. Although it is possible to obtain this power from the control circuitry through a flyback converter, the effort would be disproportionate. Instead, we allow amplified DC–DC converters, even though the available energy (wind, light) no longer matches the model scale in this case.

The main purpose of the original DC SmartGridLab was to demonstrate grid management outside of mere computer simulations. In the present AC grid testbed project, we attempted to pursue the additional goal of mimicking the dynamic behavior of equivalent full-scale grid components. We believe that we achieved this goal with the two main drawbacks of power scaling and waveform accuracy. Ideally, a microgrid simulation model would use 120 V AC to resemble a real microgrid as closely as possible; however, in a typical lab or classroom environment, the full line voltage would pose a safety hazard. In addition, the full inverter and demander power would require sizable components, such as, e.g., a 1000 W heating coil. For the purpose of this grid model, it is more practical to scale voltage and power. We used a scale factor of 1:10 over both voltage and current of residential single-phase grids. Therefore, power is scaled to 1:100. For example, the 10 W solar cell would correspond to a 1 kW solar installation at 120 V. In a trade-off for lower component costs, the waveform is not always sinusoidal. This is particularly evident in the demander (see Figure 11), but the inverter’s harmonic content increases with lower output power as well. To pursue lower harmonic distortion, additional hardware effort would be required. The demander, for example, would require a symmetrical push–pull output stage with its own isolated power supply. Similarly, minimizing the harmonic distortion of the inverter output would require feedback control of the current or—in island mode—of the voltage. A well-established approach is two-point control [29], but once again, the complexity (and thus the costs) of the power output stage would increase.

If, indeed, these factors (efficiency, 1:1 scale, waveform) are a key consideration, major semiconductor manufacturers offer reference designs and evaluation boards (see, for example, Texas Instruments application report SPRABR6 and the TIDM-SOLARUNIV micro-inverter). However, these are usually intended for full-scale grid applications, and evaluation board costs lie at least by a factor of ten above the modules presented in this manuscript. For full-scale models, control complexity can be appreciable (see, e.g., [30] for an example). On the other hand, the influence of distributed energy grids on power distribution grids has been analyzed in depth [1]. Taking the example of [1], most of the aspects covered in the review can be examined in hardware with our AC grid testbed. This includes power stability, changing topology, local power fluctuations, use of storage devices, and power flow control, to name some examples. To our knowledge, the testbed proposed in this manuscript is unique in its cost-effectiveness and its flexibility that is based on the modular design.

Moreover, we believe that we fully achieved the design goals with respect to power generation and power routing. First experiments with the integrated CORE/GridLab-D environment are promising (Figure 13). In fact, we were able to drive a 6.5 W AC load (12 V AC/6.5 W LED lamp) with solar power alone. Nonetheless, the overall conversion efficiency of the inverter system is low compared to today’s typical efficiency in full-scale systems. Both insufficient component optimization and the absence of a tight feedback control scheme are potential contributing factors. Techniques to optimize switching converters are well-known [31], but at present we decided that the efficiency is not a priority, especially in light of the amplified DC–DC converters.

The energy storage module presents a significant addition to the AC grid model in light of calls for large-scale battery storage capacity. Once again, the AC nature of the grid model significantly increases complexity over the DC SmartGridLab: in the DC SmartGridLab, energy storage is represented by a 30 farad supercapacitor, which can be discharged with up to 100 mA through a controlled current source, and it is the responsibility of the higher-level control software to select charge or discharge mode and select the current. For the AC grid, a supercapacitor would impose limitations similar to those of the wind turbine (only 5 V at less than 100 mA), and we decided to use 18,650-type lithium ion batteries. Their combined capacity of approximately 30 Wh fits the overall scaled power levels, with the equivalent full-scale installation being a 3 kWh battery bank. In its present configuration, however, no higher-level control is provided. If AC grid power is present, the DC–DC converter is disabled and the associated inverter is powered down. At the same time, the charging circuit is activated. When the AC utility fails, the DC–DC converter is activated, and the inverter begins to supply power from the batteries. At present, the storage module requires a grid–subgrid configuration. Otherwise, the storage module would constantly switch between charging and discharging modes as it senses its own power on the grid. In a future revision, we will combine storage and inverter into one module and make a simple software modification that allows high-level control of its operation.

Concluding, we designed the modules for an AC grid testbed, which fulfill a similar functionality to the underlying DC testbed modules [19] that we used as an orientation: we can introduce varying levels of renewable energy into the grid, draw controllable amounts of energy, route power between subgrids by means of bridges, and measure the power at any point of the grid. We also integrated the modules on the software level with the CORE-based emulator that we are developing in parallel. Moving beyond the DC testbed, we have the ability to introduce reactive loads to measure active, apparent, and reactive power and the power factor. We can also measure the waveform for advanced computations such as harmonic content and harmonic distortion. A few areas of improvement remain, but we aim to collect some practical experience before deciding to what extent the shortcomings need to be addressed. The AC grid modules can be considered low-cost, because the component costs are in a range comparable to the DC grid testbed modules.

In addition, the AC testbed modules offer several new features that we have not explored yet. For example, the demander can be programmed to draw current with an arbitrary waveform and can emulate, for example, a non-PFC corrected switch mode power supply. The two-stage inverters can be connected to multiple sources via their DC-link, which would create a DC bus similar to those found in microgrid installations. With additional practical experience, we expect to successively explore those additional options.

Lastly, the modules presented here constitute downscaled models for the basic modules of a CERTS microgrid [32], and most of the CERTS functionality exists. Most notably, the modules connect to the grid in a peer-to-peer fashion without a master controller, and all modules have their own, localized control system. The higher-tier micro-PC is optional. Modules have their own fault detection; for example, the bridge module automatically disconnects and places a subgrid into island mode when a fault is detected. The only exception is the full implementation of droop control, although the use of a defined phase shift in the inverter can widely emulate the effects of droop control, notably the introduction of reactive power.

## Figures and Tables

**Figure 1 sensors-23-03235-f001:**
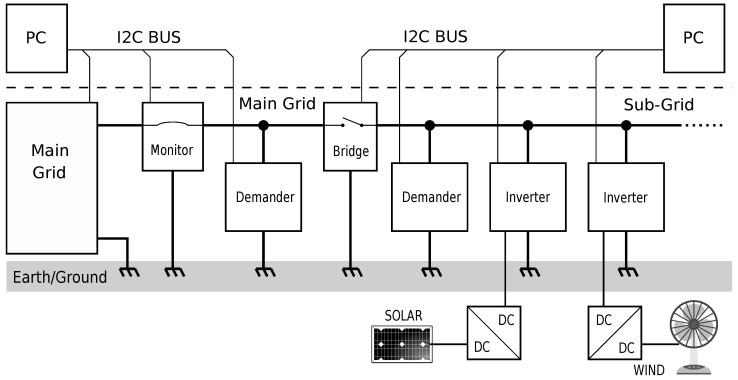
Overview of the AC grid modules in a possible grid/subgrid configuration. A transformer emulates the stiff bus bar of the utility main grid, and a monitor measures the power drawn. A grid-to-grid bridge unit separates the main grid from a microgrid or subgrid and ensures that a connection is only established when the two grids are in phase. Inverters feed energy from local sources (solar, wind) into the grid, and demanders provide a programmable load. Higher-level control tiers are implemented in micro-PCs that are connected with the low-level control tier via an I2C bus.

**Figure 2 sensors-23-03235-f002:**
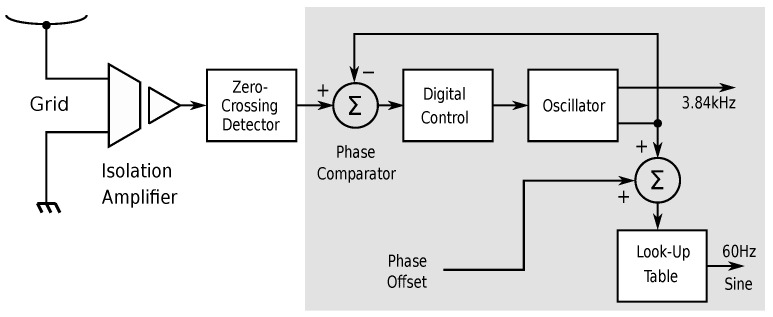
Schematic representation of the PLL. The AC voltage from the grid is fed into a zero-crossing detector. At each zero crossing, the AC grid phase is compared to that of the internal oscillator. The phase error is used to adjust the oscillator frequency so that the phase error is driven to zero and the PLL therefore locks. The internal oscillator is used to derive the sampling frequency for the sine wave generation and for sampling the grid voltage and current. Components inside the gray-shaded box are realized in software.

**Figure 3 sensors-23-03235-f003:**
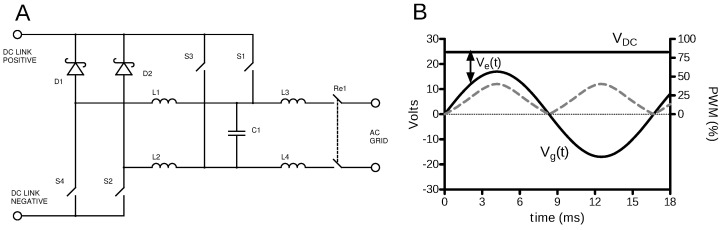
Principle schematic of the inverter power stage (**A**) and associated waveforms (**B**). Two half-bridges are formed by S1, D2, S2, L2, and S3, D1, S4, L1, respectively. Each half-bridge forms a switch-mode buck converter, and from a frequency-domain perspective, the output stage is formed by a three-pole LCL lowpass filter. Due to the buck configuration, the DC-link voltage $V_{DC}$ must exceed the peak grid voltage ${\hat{V}}_{g}$ as indicated in (**B**). From the voltage excess $V_{e}=V_{DC}-\left|V_{g}\right|$, the pulse width (dashed gray line) can be computed.

**Figure 4 sensors-23-03235-f004:**
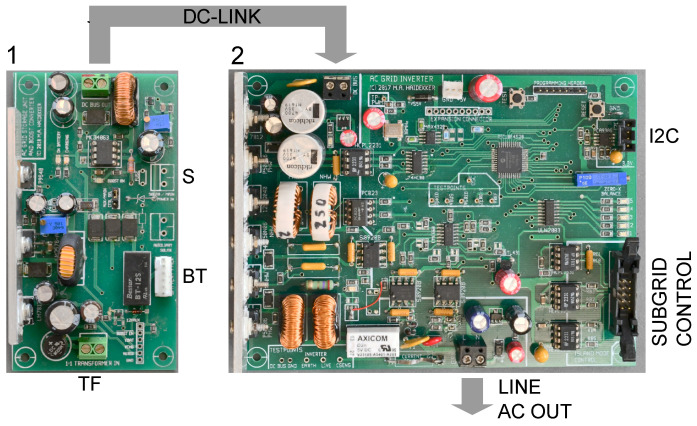
Photos of the DC-link boost converter (1) and the inverter (2). The different sections of the inverter are recognizable: H-bridge with LCL filter and DC-link bypass capacitors on the left side, with the control component using most of the right part of the PCB. The subgrid control input allows connection to a bridge, which—if connected—takes control over the inverter’s frequency and phase. The DC-link is fed by a boost converter (1), which can be configured in various ways: Power from a solar cell or wind turbine (S) are fed directly into the boost converter. Alternatively, power from a main grid or transformer (TF) can be used to charge the battery (BT), which, upon power failure at TF, is connected to the converter and, via the DC-link, activates the inverter. A third connection (not labeled) allows to use the signal from a weak solar cell or wind turbine to control power draw from a lab power supply.

**Figure 5 sensors-23-03235-f005:**
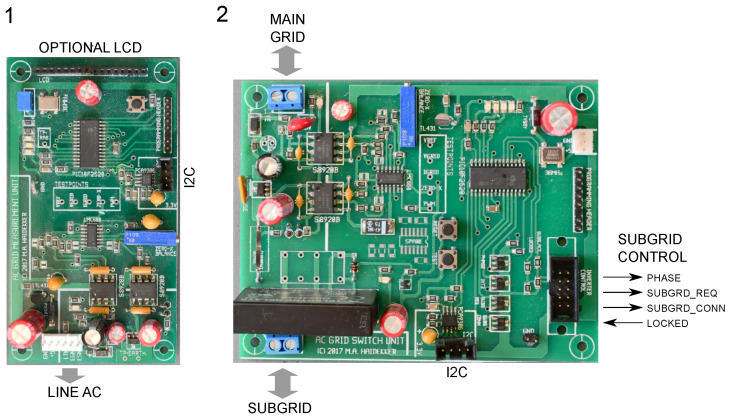
Photos of two representative modules: (1) grid monitor and (2) grid-to-grid bridge. The demander is similar to the monitor with only an added controllable load in the AC section. All modules have I2C connectivity for the second-tier control that takes place on a BeagleBone Black micro-PC. The grid monitor measures grid voltage and and power delivery. It can collect V/I samples over one full AC cycle, and it can monitor voltage, current, power, power factor, and frequency. These can be recorded with the micro-PC or can be observed in real time with the optional LCD. The grid-to-grid bridge shares some of the functionality of the monitor, but its main purpose is to create the subgrid control signals to synchronize any attached inverters (Figure 4). When all inverters signal phase lock, the solid-state relay (near the subgrid connector) establishes the bridge connection. For this purpose, four signal lines exist that establish direct subgrid control of the inverters.

**Figure 6 sensors-23-03235-f006:**
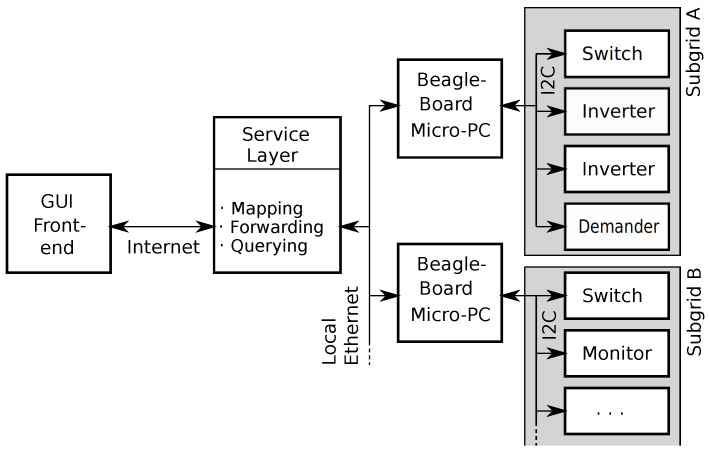
Communication of the AC grid hardware with CORE. Intermediate-level control is performed by BeagleBone micro-PCs, who communicate with one or more grid modules through an I2C bus. High-level communication and detailed grid information is communicated with the CORE server through the local Ethernet.

**Figure 7 sensors-23-03235-f007:**
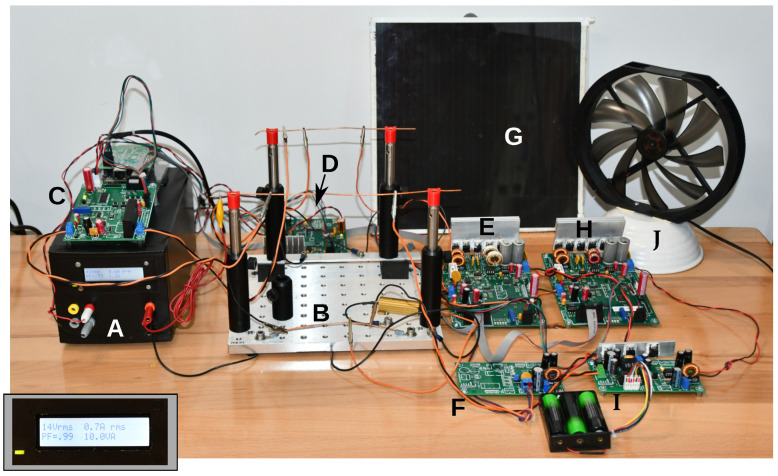
Demonstration grid setup, similar to the one sketched in Figure 1. The utility grid is represented by the transformer (**A**). Inset: Inside the transformer box is a grid monitor that displays the power conditions in real time. The utility AC voltage is delivered to the power grid model (**B**), which contains an aluminum plate (earth) and two wires (live; utility grid and subgrid). The two wires can be connected by means of the bridge (**C**). A demander (**D**) is found in the rear. Two inverter groups are connected to the subgrid: inverter (**E**) with DC-link converter (**F**) and solar cell (**G**), and inverter (**H**) with storage unit (**I**) and a battery pack. A wind wheel (**J**) is included for demonstration, but not connected. Wires include not only the AC line wires, but also the I2C bus and the 5 V supply to the modules.

**Figure 8 sensors-23-03235-f008:**
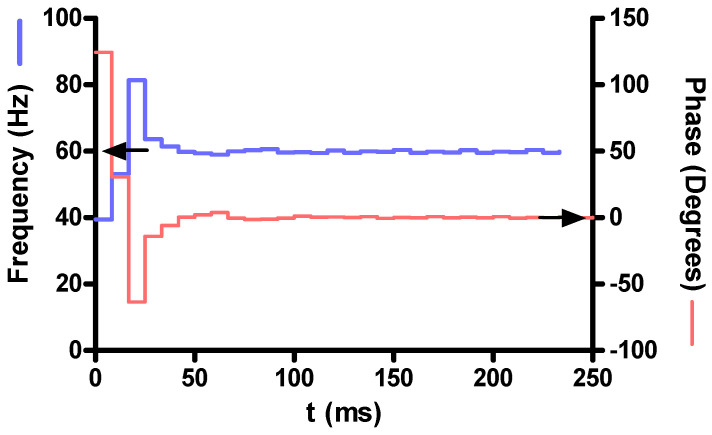
PLL synchronization behavior in an example of extreme frequency mismatch. The PLL was set to run at 40 Hz before the 60 Hz AC signal was applied. The PLL settled into a tolerance band of 0.005 π within 120 ms. Arrows indicate the steady-state frequency of 60 Hz and phase of 0°.

**Figure 9 sensors-23-03235-f009:**
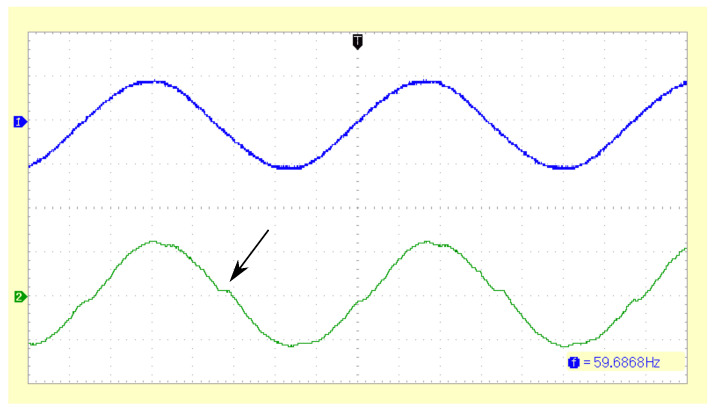
Inverter output waveform. The upper trace shows the AC grid voltage (20 V/div). The inverter supplies 5 W into the grid, and its output current is measured across a 0.1 Ω shunt resistor and is shown in the lower trace (0.5 A/div). Some deviations from the ideal sinusoidal waveform are visible, notably near the zero crossings (arrow), where integer rounding errors lead to a pulse width of zero.

**Figure 10 sensors-23-03235-f010:**
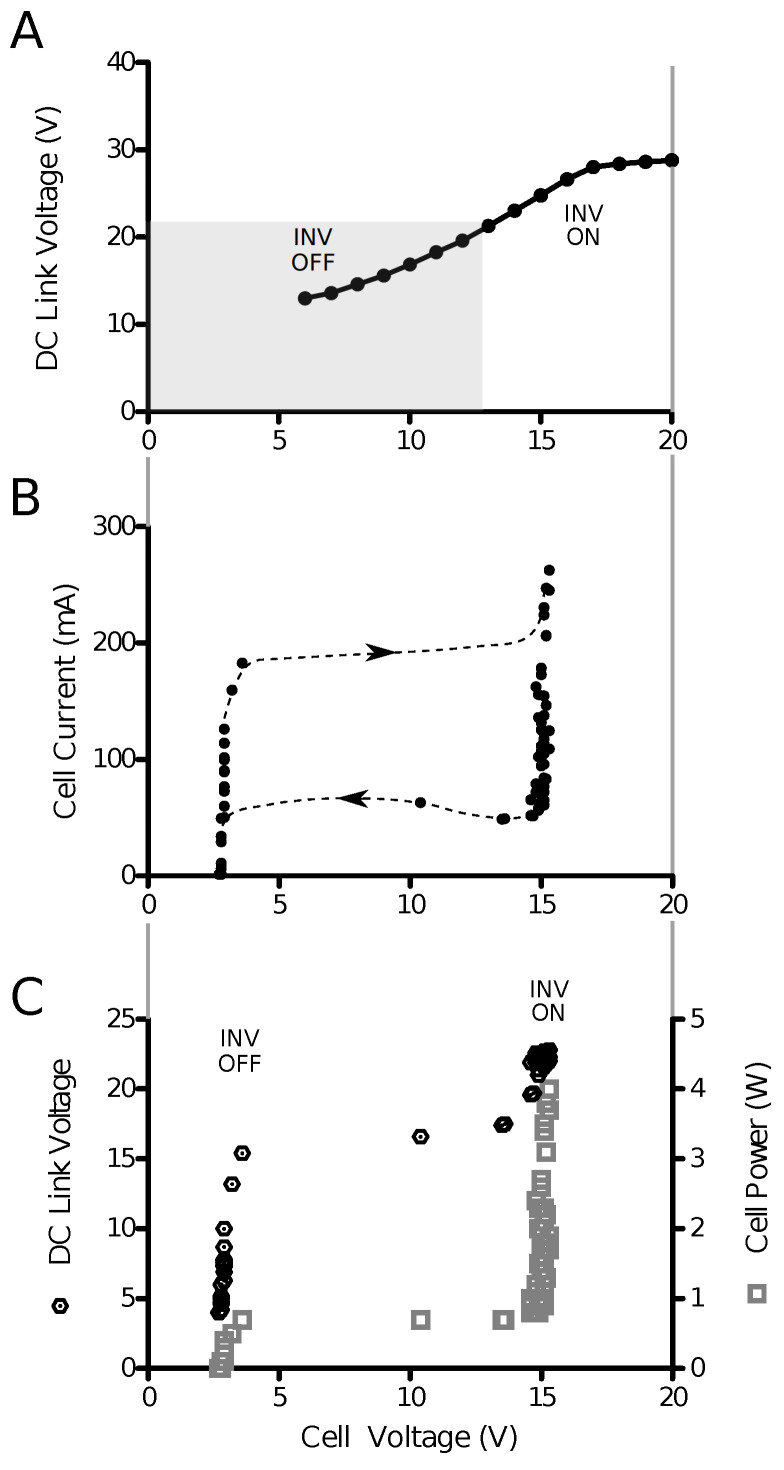
Power tracking behavior of an inverter with DC–DC converter and solar cell. (**A**) Voltage transfer function of the DC–DC converter. The inverter turns on its H-bridge when the curve leaves the gray shaded box, which ensures that enough solar irradiation is present to maintain power transfer to the grid. (**B**) V–I characteristic of the power tracking scheme with respect to the solar cell. As soon as the solar cell’s output power rises above a certain threshold level, the inverter begins drawing power. Because of the proportional relationship between the DC-link excess voltage and the current, the inverter is allowed to reduce its current demand significantly before it is forced to turn off the bridge, and the result is a hysteresis-like behavior (dashed lines). The hysteresis lines were added manually and are only approximate. The behavior is complex, because the boost converter requires a minimum voltage to operate, which is much lower than the inverter turn-on voltage. (**C**) When sufficient solar irradiation is present, the output power is predominantly determined by the solar cell output current, and the cell voltage is maintained near its power maximum.

**Figure 11 sensors-23-03235-f011:**
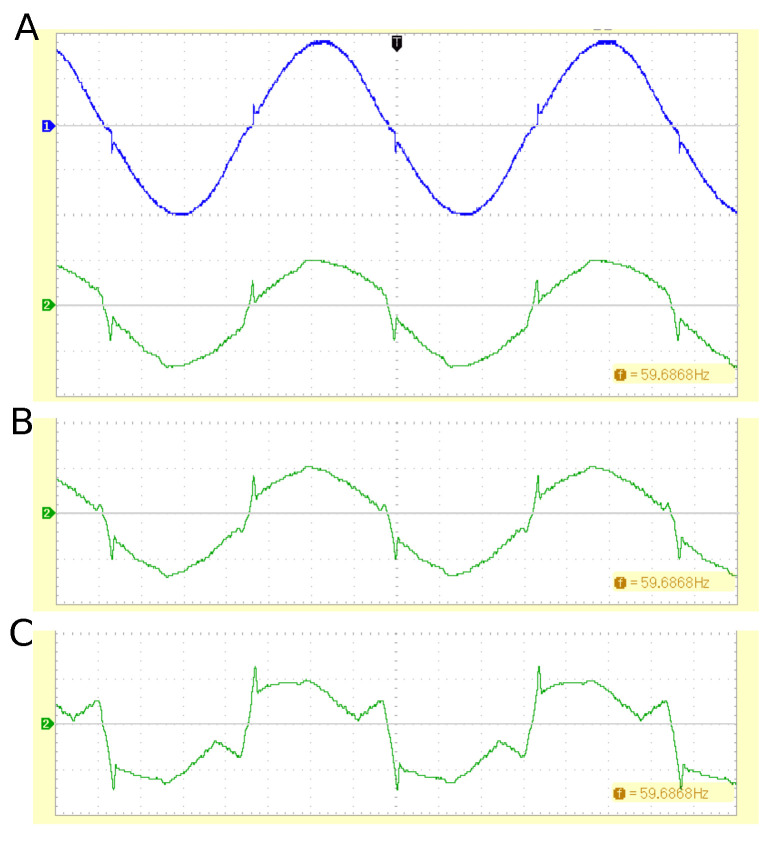
Representative examples of demander waveforms. (**A**) The top trace shows the grid voltage (10 V/div), and the bottom trace shows the current drawn (0.5 A/div) for an emulated resistive load. Spikes occur immediately after the zero crossing when the operational amplifier’s negative input voltage slightly lags the control voltage and the output rises steeply. (**B**) Current waveform for 45° phase (leading). (**C**) Current waveform for 90° phase (leading). The zero-crossing behavior of the op-amp combined with the absence of a true reactive component causes a highly distorted waveform.

**Figure 12 sensors-23-03235-f012:**
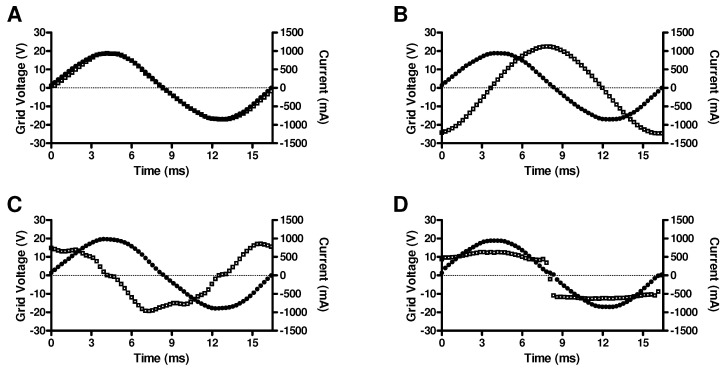
Discrete sampled grid voltage $V_{k}$ and current $I_{k}$ for different types of load: (**A**) resistor of 20 Ω; (**B**) inductor of 31 mH; (**C**) capacitor of 120 μF; (**D**) LED lamp with built-in rectifier and inverter. Discrete data points are shown as glyphs (filled circles: voltage; open squares: current) to highlight the sampling frequency in relationship to the AC grid frequency.

**Figure 13 sensors-23-03235-f013:**
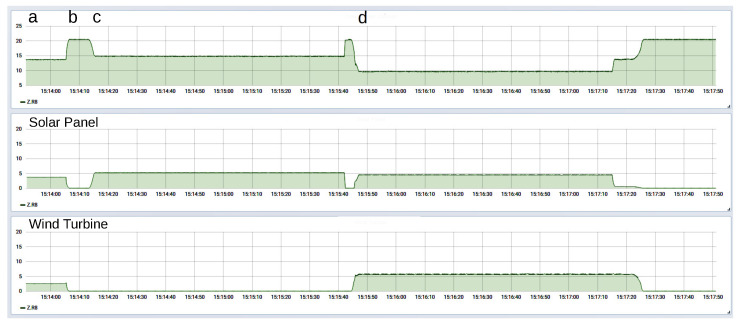
Power distribution with two intermittent power sources, collected from AC grid hardware modules and visualized with Grafana. Initially (time point a), both renewable sources are active and the transformer (i.e., the larger utility grid) supplies only the difference. When one or both renewable sources fail (b, c), the transformer power output rises to compensate for the shortfall, but returns to lower levels when renewable energy is again supplied (d).

## Data Availability

The data presented in this study, including schematics and firmware code, are available on request from the corresponding author. The data are not publicly available due to their volume and complexity.

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
