# Peer review of "Alternating-Current Microgrid Testbed Built with Low-Cost Modular Hardware"

_sensors, 2023, doi:10.3390/s23063235_

Round 1

Reviewer 1 Report

It isn't very sure to publish this article in SENSORS - a little about sense, more about grid.

1) References are too old - from 2010 to 2016;

2) Typically ∑ is from value -to value, here from 1 to N - expressions (3); 

3) Zero crossing detection still is a weak point today due to noise, harmonics, grid frequency light sweep and connected non-linear consumers (without power factor correction). Many articles and algorithms propose solutions to this topic in literature sources and over the internet (Static Energy Meter Errors Caused by Conducted Electromagnetic Interference, 2016 IEEE Electromagnetic Compatibility Magazine – Volume 5 – Quarter 4, for example);

4) Scaling is not the best solution to imitate the situation - 120 V AC is quite different from 12 V AC in practice;

5) article is wordy in a bit. Use some tables where it is possible to reduce size.

Reviewer 2 Report

This paper is well presented. I have some comments as following:

1.     Please shorten the current Abstract and give a brief description about the motivation, significance and contribution of your research.

2.     In Section I, please try to improve the first and second paragraphs as the description for microgrid where the proposed method is applied is limited.

3.     Literature review can be improved with more studies, such as Predictive voltage hierarchical controller design for islanded microgrids under limited communication.

4.     Please highlight the contribution part in Section I that cannot fully reflect the main contributions and advantages of the proposed method.

5.     The authors must give more discussion about controller design and method implementation using the proposed method. At the same time, it is difficult to find how to design the proposed control system in microgrids, so please give a figure with detailed controller design.

6.     At present, there are only three formulas, which are often insufficient for theoretical research with a clear background. Therefore, the theoretical analysis should be strengthened, and more formula derivation and performance analysis should be given

7.     The authors are suggested to give more comparison results with other representative methods, highlighting the main advantages and better performance of the proposed method.

Round 2

Reviewer 1 Report

The title is "Alternating-Current Microgrid Testbed Built With Low-Cost Modular Hardware".

Regarding hardware, everything seems good, with nice breadboards.

Doubtful is I2C communication - it could be too slow. Besides, according to I2C-bus specification and user manual, practically limited to a max of 10 metres at a clock frequency of 10 kHz for reliable data communication. Suitable for Testbed, but not usable out into the field. 

Still, the whole idea, except the testbed hardware, is difficult to understand:

a) the two AC grids to be connected aren't well described; one is a standard AC grid, and the second is an islanded AC grid. In islanded AC grid, one is the central inverter setting the frequency, and the other inverters are slaves regarding AC frequency.

b) to connect to the primary AC grid (here created by transformer), all inverters must be frequency tuned up on the isolated AC grid.

All inverters on the isolated AC grid may keep the frequency and phase close to the primary AC grid utilizing communication.

After connecting both grids, the inverters become grid-tied inverters. What occurs near the grid's connection moment- not described, and many other points too.

c) Don't show some measurements for 1W, 2W, or 3W power due to possible inaccurate measurements achieved by 10-bit ADC without pointing to the shunt resistor value, the amplifier amplification coefficient and noise characteristics.

It is not acceptable to publish in Sensors without general sensing parameters.

So this could not be the genuine Testbed for grid imitation or simulation.

In general - no reason to build something without real practical value.

Finally - no reason to discuss the technical design of the test bed without discussing actual implementation.
